

# Development and Validation of a New Ozone Dataset Using Complete Data Fusion of MIPAS and IASI Observations: A Step Towards Understanding Stratospheric Ozone Intrusions in the Himalayan Region

Liliana Guidetti[1,2], Erika Brattich[1], Simone Ceccherini[2], Michaela I. Hegglin[3], Piera Raspollini[2], Cecilia Tirelli[2], Nicola Zoppetti[2], and Ugo Cortesi[2]

[1]Università di Bologna, Dipartimento di Fisica e Astronomia (DIFA) "Augusto Righi", via Irnerio 46, 40126 Bologna, Italy
[2]Istituto di Fisica Applicata "Nello Carrara" del Consiglio Nazionale delle Ricerche, Via Madonna del Piano 10, 50019 Sesto Fiorentino, Italy
[3]Institute of Climate and Energy Systems – Stratosphere (ICE-4), Forschungszentrum Jülich GmbH, Wilhelm-Johnen-Straße, 52428 Jülich, Germany

**Correspondence:** Liliana Guidetti (liliana.guidetti3@unibo.it) and Nicola Zoppetti (n.zoppetti@ifac.cnr.it)

**Abstract.** We present a new ozone dataset generated using the Complete Data Fusion (CDF) algorithm, which combines limb observations from MIPAS-ENVISAT along with nadir observations from IASI during their overlapping operational period (2008–2011). The fusion aims to enhance ozone profile information in the Upper Troposphere–Lower Stratosphere (UTLS), particularly over the Himalayas—a key region for stratospheric intrusion events. The fused dataset, mapped onto the IASI-
MetOp grid, includes ozone partial columns with associated uncertainty estimates (covariance matrices, averaging kernels, and a priori profiles). After tuning the algorithm on 2008 data, the product was validated for the 2009–2011 period against ozonesondes from WOUDC stations across multiple latitude bands. Results show that the fused profiles improve vertical information content (degrees of freedom) and reduce uncertainty, successfully propagating MIPAS information to lower altitudes even in regions lacking direct limb observations. While the dataset initially exhibited altitude- and latitude-dependent biases, primarily inherited from IASI, these were significantly reduced through bias correction. The final product is suitable for data as-
similation and model evaluation, offering valuable support for atmospheric reanalyses and studies of troposphere–stratosphere exchange processes.

## 1 Introduction

Tropospheric ozone is a regulated air pollutant and a greenhouse gas with a central role in chemical, dynamical, and radiative
processes in the atmosphere. Changes in its concentrations affect the climate through radiative and biogenic feedbacks and pose risks to human and ecosystem health (Fuhrer et al., 1997; Joiner et al., 2009; Fleming et al., 2018). Although it has no direct emission sources, it is produced through photochemical reactions involving sunlight, volatile organic compounds (VOCs), and nitrogen oxides (NOx) emitted by natural and primarily anthropogenic activities (Crutzen and Lelieveld, 2001; Zhang et al.,





2019; Donzelli and Suarez-Varela, 2024). Its short lifetime, spatial variability of sources and sinks, and dependence on sunlight
result in significant spatial and temporal variability (Tarasick et al., 2010; Prather and Zhu, 2024).

A key natural source of tropospheric ozone is atmospheric circulation (Cristofanelli et al., 2015), particularly through Stratospheric-Tropospheric Exchange (STE) processes, which facilitate the bidirectional transfer of air masses, gases, and aerosols between the stratosphere and troposphere. STE dynamics, driven by mechanisms like Brewer-Dobson circulation, tropopause folding, and atmospheric waves, significantly influence tropospheric ozone concentrations (Holton et al., 1995; Stohl et al., 2003; Mo-
hanakumar, 2008; Fueglistaler et al., 2009; Gettelman et al., 2011). Stratospheric contributions to tropospheric ozone account for 25-50% in the extratropics, with the strongest influence in the Southern Hemisphere (Banerjee et al., 2016). Low strato-spheric ozone explains up to 70% of the variability in mid-tropospheric ozone at mid-latitudes (Hess and Zbinden, 2013). A global overall statistically significant increase in tropospheric ozone has been reported (Williams et al., 2019), and mainly attributed to changes in atmospheric circulation and the strengthening of the Brewer-Dobson Circulation (BDC) (Hegglin and
Shepherd, 2009; Butchart, 2014; WMO, 2018).

The Himalayan region is particularly affected by stratospheric intrusions due to its high altitude, complex topography, and dynamic meteorological conditions (Stohl et al., 2003; Randel and Park, 2006). Processes such as tropopause folding, mon-soonal convection, and interactions with the subtropical jet stream facilitate the downward transport of ozone-rich air masses. Studies have reported significant stratospheric contributions to surface-level ozone in this region, with a 25% contribution de-
tected during winter (Cristofanelli et al., 2010). The rise in tropospheric ozone levels near the southern flank of the Himalayas has also been associated with increased stratospheric influence (Williams et al., 2019). Despite these observations, substantial uncertainty remains in explaining surface ozone variations due to discrepancies between models and observations.

Our current understanding of the spatial and temporal climatology of tropospheric ozone is closely related to our ability to mea-sure and simulate it. In this context, determining the stratosphere-to-troposphere exchange (STE) contribution to tropospheric
ozone concentrations is crucial. Ozone observations are currently performed using ozonesondes and other in-situ and remote sensing instruments. These measurements are often combined with statistical models based on ozone tracers (e.g. Be, Pb, Rn)(Brattich et al., 2017, 2021). While in-situ measurements are limited in spatial and temporal coverage, remote sensing ob-servations offer a broader perspective, despite the constraints introduced by the inversion techniques required for their retrieval. Dynamical and chemical models are also widely used to simulate STE events and are often compared with observations; how-
ever, systematic biases between models and observations still exist (Hegglin et al., 2010; Williams et al., 2019). Understanding STE processes, particularly in regions such as the Upper Troposphere–Lower Stratosphere (UTLS), requires atmospheric ob-servations with both high vertical resolution—due to the strong vertical gradients—and adequate horizontal resolution, since these events often occur over spatially confined regions. Satellite remote sensing instruments offer complementary strengths: limb-sounding sensors provide high vertical resolution but limited horizontal coverage, while nadir-viewing sensors offer dense
horizontal sampling but lower vertical resolution. Combining coincident measurements from both observation geometries can help to produce ozone profiles that retain fine vertical structure while improving horizontal representation.

In this context, we combine limb observations from the Michelson Interferometer for Passive Atmospheric Sounding (MI-PAS), operating aboard ESA's Envisat between 2002 and 2012 (Fischer and Oelhaf, 1996; Fischer et al., 2008), with nadir



observations from the Infrared Atmospheric Sounding Interferometer (IASI), active since 2006 aboard EUMETSAT's MetOp
satellite series (Clerbaux et al., 2009; Razavi et al., 2009; Coheur et al., 2009). Both instruments operate in the mid-infrared
(MIR) spectral range and provide valuable, complementary information on atmospheric composition. To optimally exploit their
synergy, we apply the a posteriori Complete Data Fusion (CDF) methodology (Ceccherini et al., 2015), which has previously
been used to merge nadir–nadir simulated measurements (Zoppetti et al., 2021; Tirelli et al., 2023), and limb-nadir simulated
measurements (Tirelli et al., 2025), producing combined datasets with improved vertical information content. Here, we apply
the latest version of the CDF method (Ceccherini et al., 2022) to real measurements.

In this paper, we present the development and validation of a novel 2008-2011 ozone dataset, created using the CDF method-
ology applied to MIPAS and IASI observations, and assess its quality using ozonesonde data. The Himalayan region is used
as a reference and it is the focus area in part of the analysis due to its relevance for STE processes, which motivate this study.
The main objective of this work is to evaluate and enhance the downward propagation of information from high-quality limb
measurements in the stratosphere to the troposphere. In this lower region, MIPAS sensitivity decreases, retrieval uncertain-
ties increase, or MIPAS data may be missing, whereas IASI provides more consistent coverage. This approach leverages the
ability of MIPAS to resolve the UTLS (Upper Troposphere-Lower Stratosphere) region and the broader horizontal coverage
and near-surface sensitivity of IASI. Notably, Ceccherini et al., 2010 previously fused IASI and MIPAS data using the Mea-
surement Space Solution (MSS) method (Ceccherini et al., 2009), which was later shown to be equivalent to the CDF method
(Ceccherini, 2016), demonstrating the benefits of combining measurements from these two sensors.

The paper is structured as follows: Section 2 outlines the data and methods, including descriptions of the input datasets, a
summary of the fundamental principles behind the CDF approach, algorithm choices, and the validation strategy. Section
3 presents the resulting dataset and discusses its validation and inter-comparison. Finally, Section 4 summarizes the main
conclusions of this work.

## 75   2   Data and Methods

### 2.1   Data

#### 2.1.1   MIPAS Data

The Michelson Interferometer for Passive Atmospheric Sounding (MIPAS) was launched aboard ESA's Envisat (2002–2012)(Fis-
cher and Oelhaf, 1996). Operating in a Sun-synchronous orbit at an altitude of approximately 800 km, it allowed dense vertical
($\sim$3 km) and horizontal ($\sim$500 km) coverage, providing data relevant for the study of chemical and dynamical atmospheric
processes, and for stratospheric-tropospheric interactions. By retrieving vertical profiles of several atmospheric trace species,
MIPAS addressed diverse scientific objectives, especially in the UTLS (Chauhan et al., 2009; Höpfner et al., 2015; Sembhi
et al., 2012). Despite orbit adjustments in 2010, the mission continued until Envisat's abrupt loss in April 2012.

The instrument was a Fourier transform spectrometer designed to detect limb emission spectra from the middle and upper
atmosphere (Fischer et al., 2008). Operating in the mid-infrared range (from 4.15 $\mu$m to 14.6 $\mu$m, or 2410 - 685 cm$^{-1}$), MIPAS





provided high spectral resolution. Initially, MIPAS operated at full resolution (FR) with a step size of up to 0.025 cm$^{-1}$, and with 17 limb views per scan (covering altitudes from 6 to 68 km), providing a vertical sampling of approximately 3 km in the troposphere and 3-8 km in the stratosphere. Following technical issues in 2004, MIPAS switched to optimized resolution (OR) in 2005 with a step size of up to 0.0625 cm$^{-1}$, increasing the number of views to 27 per scan (covering altitudes from 3 to 70

km) and allowing a vertical sampling of about 1.5 km. MIPAS maintained a 100% duty cycle during FR but initially dropped to 30% in OR, gradually recovering to full operation by 2007. Horizontally, the instrument had a field of view of about 30 km, with a sampling distance of nearly 500 km along the orbit, providing 72-75 vertical profiles per orbit (Dinelli et al., 2021).

Level 2 (L2) products from MIPAS spectral measurements serve as input for the CDF algorithm. Retrieval of these products is performed using the Optimised Retrieval Model (ORM) v8.22, which is the Level 2 processor developed for the final

full-mission reanalysis of MIPAS data (Raspollini et al., 2022). The retrieval process follows an iterative approach where the difference between the radiance computed by the forward model for a given atmospheric state and the observations is minimised, using the Jacobian matrix, describing the sensitivity of the radiance to variations in the retrieved parameters. Regularization techniques are applied to stabilize the inversion and reduce uncertainties. The operational dataset includes vertical profiles of temperature, pressure, and 21 trace gases, such as H$_2$O, O$_3$, HNO$_3$, CH$_4$, N$_2$O, NO$_2$, CFC-11, ClONO$_2$, N$_2$O$_5$,

CFC-12, COF$_2$, CCl$_4$, CF$_4$, HCFC-22, C$_2$H$_2$, CH$_3$Cl, COCl$_2$, C$_2$H$_6$, OCS, and HDO. The retrieval methodology accounts for different observational phases of the mission, including Full Resolution (FR) (2002–2004) and Optimized Resolution (OR) (2005–2012), ensuring consistency across the entire dataset (Dinelli et al., 2021).

In this study, we reprocessed ozone MIPAS OR measurements from 2008 to 2012 using ORM v8.22, configured with the OE method as required by the CDF framework. The initial guess consists on the IG2 as described Remedios et al. (2007), then

the algorithm considers values of the previous scan as described in Raspollini et al. (2022). The inputs for the CDF algorithm includes ozone state vectors, variance-covariance matrices (VCMs), averaging kernels (AKs), and the a priori information for each product, the a priori profiles. State vectors are composed of volume mixing ratio (VMR) ozone profiles, water continuum and offsets values. The reprocessed MIPAS ozone dataset has been validated to ensure its scientific reliability.

### 2.1.2  IASI Data

The Infrared Atmospheric Sounding Interferometer (IASI) is a multi-purpose sounding instrument composed of a Fourier transform spectrometer based on Michelson inteferometer (Blumstein et al., 2004). IASI has been providing continuous data since October 2006, when it was first launched on board MetOp-A. It was followed by IASI-B (MetOp-B), launched in 2012 and IASI-C (MetOp-C), launched in 2018. The three instruments (IASI -A, -B and –C) cover the spectral range 2760 - 645 cm$^{-1}$, with a spectral sampling of 0.25 cm$^{-1}$, for a total of 8461 spectral channels. IASI is a nadir viewing instrument, that

scans the atmosphere symmetrically across-track with respect to the nadir direction in 30 scan steps, for a total swath width of ± 1100 km. The Field of Regard (FOR) of IASI comprises a 2x2 matrix of circular samples Instantaneous Field of View (IFOV). At the nadir direction, and a satellite altitude of 819 km, each IFOV corresponds to a ground resolution of approximately 12 km, although this size increases varying with the scan angle (ellipse-shaped). Each FOR is centered on the viewing direction and corresponds to a footprint of about 50 km x 50 km at the nadir. The instrument scans in a step and stare mode and is able to





acquire a total of 30 x 120 interferograms per scan line. IASI delivers vertically resolved atmospheric profiles of temperature and humidity and retrieves the concentrations of key greenhouse gases. Its data are critical for a wide range of applications, including improving numerical weather prediction (NWP), monitoring atmospheric composition, assessing air quality, and studying long-term climate variability (Clerbaux et al., 2009; Razavi et al., 2009; Coheur et al., 2009). Additionally, IASI has been widely recognized for its capability to provide accurate radiance measurements in clear-sky conditions, making it a reference dataset for validating and evaluating climate models.

The ozone data used in this study come from the OZOS IASI-A archive, available on the IASI AERIS Data Portal. This dataset was generated using the Fast Optimal Retrievals on Layers for IASI (FORLI-$O_3$) processor. The FORLI processor is based on the OE (Optimal Estimation) method to derive atmospheric ozone profiles from IASI spectral data in the thermal infrared region (1000–1070 cm $^{-1}$), using the ozone climatology by McPeters et al. (2007) as a priori information (Hurtmans et al., 2012; Boynard et al., 2016). The operational dataset provides vertical ozone profiles in partial column density units (Dobson Units, DU) on 40 layers between surface and 40 km, with an extra layer from 40 to 60 km, the top of the atmosphere. In addition to ozone profiles, the product includes key diagnostic information, such as VCMs, AKs, and a priori partial column profile. However, the a priori VCM is provided separately and must be reconstructed as described in the Product User Manual (PUM) for Near real-time IASI total $O_3$ and $O_3$ profile. FORLI $O_3$ data have been available since 2008; in this study, we used data corresponding to the MIPAS activity period (2008–2012).

### 2.1.3 Ozonosondes Data

The ozonesonde data used in this study to validate the newly developed ozone dataset were obtained from the World Ozone and Ultraviolet Radiation Data Centre (WOUDC), which collects and archives ozone profile measurements from ozonesondes launched at various locations worldwide. These data are publicly available in the WOUDC archive and can be downloaded by following the instructions provided on the Data Access webpage.

A total of 33 ozonesonde stations were selected for this analysis based on their spatiotemporal coincidence with the fused ozone profiles and the individual MIPAS and IASI datasets. These stations are highlighted in red in Figure 1. As shown, the distribution is primarily concentrated between 30°N and 60°N, with a higher density over Europe and the United States.

## 2.2 Methods

### 2.2.1 Complete Data Fusion

This section, devoted to the description of the CDF method, is divided into three subsections. The first section briefly presents the CDF formulation, the second describes the CDF setup, and the third focuses on CDF tuning carried out in this study. To reserve the 2009–2011 period for validation, both setup and tuning were performed using 2008 data only. These two steps are presented separately to clearly distinguish the setup phase—essential for CDF and closely tied to input data quality—from the tuning phase, which can be a strength of the method by allowing parameter adjustments.





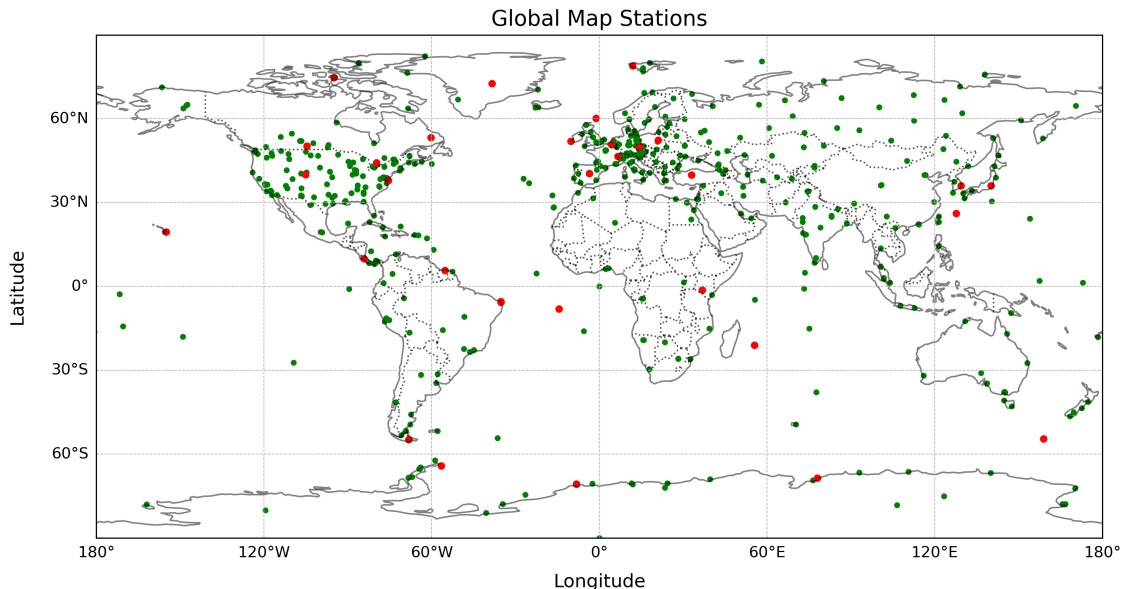

**Figure 1.** Global map of WOUDC stations (green markers). Red markers identify stations effectively used in this study.

## CDF Formulation and Diagnostics

Complete Data Fusion (CDF) is a technique developed by Ceccherini et al., 2015 that merges independent profile measurements from different data sources into a unified dataset. By integrating complementary datasets, CDF maximizes the information content while minimizing the uncertainties associated with individual measurements. Under the assumptions of a linear forward model and perfectly matching measurements, the results obtained with CDF are identical to those of synergistic retrievals, but with significantly lower computational costs. The method has been progressively refined through various improvements, culminating in a generalized formulation in 2022 (Ceccherini et al., 2022). The technique has been widely applied to simulated datasets from different sensors, consistently outperforming the input products (Zoppetti et al., 2021; Tirelli et al., 2021; Ridolfi et al., 2022).

In this work, we applied the generalized CDF formula to spatially and temporally coincident IASI and MIPAS L2 profile pairs. Coincidence was defined by a maximum separation of 200 km and 1 hour, criteria chosen to match the sampling characteristics of MIPAS and IASI and to suit applications such as stratospheric intrusion studies. Because MIPAS has coarser spatial sampling than IASI, we centered the fusion on the IASI profiles. As a result, each MIPAS profile can be fused multiple times with several nearby IASI profiles. This strategy is non-trivial: it is designed to preserve the spatial density of IASI while still leveraging the vertical information content of MIPAS. This approach has both benefits and drawbacks. On the one hand, it enhances the spatial representativity of the fused dataset, allowing for denser spatial sampling around MIPAS tracks. On the other hand, it introduces some redundancy in the stratospheric contribution from MIPAS, which is replicated across multiple fused profiles. Nonetheless, given the scientific objectives of this work, this choice remains well justified.





Due to the non-coincident nature of the profiles, we adopted the generalized formulation of the CDF method, which accounts for both interpolation and coincidence errors (Ceccherini et al., 2018, 2022). For clarity, we report here the main equations in the general case. Let us assume to have $N$ independent retrieved vertical profile ($\hat{x}_i$; $i=1,..., N$), each defined on different vertical grids, with varying vertical coverage and referring to different locations in space and time.

The fused profile $x_j$ can be computed as:

$$x_{\mathbf{f}} = \Big( \sum_{i=1}^{N} \mathbf{H}_i^{\#T} \tilde{\mathbf{S}}_i^{-1} \mathbf{A}_i \mathbf{H}_i^{\#} + \mathbf{S_a}^{-1} \Big)^{-1} \Big( \sum_{i=1}^{N} \mathbf{H}_i^{\#T} \tilde{\mathbf{S}}_i^{-1} \tilde{\boldsymbol{\alpha}}_i + \mathbf{S_a}^{-1} x_{\mathbf{a}} \Big), \tag{1}$$

where $\mathbf{A}_i$, $\mathbf{S}_a$, $x_a$, are respectively the averaging kernel matrix (AK) of each fused profile, and the chosen a priori covariance matrix (CM) and profile for the fusion. $\mathbf{H}_i^{\#}$ is the generalized inverse matrix of the interpolation matrix $\mathbf{H}_i$, which interpolates profiles from the retrieval grid to the fusion grid. $\tilde{\boldsymbol{\alpha}}_i$ and $\tilde{\mathbf{S}}_i$ are respectively defined as:

$$\tilde{\boldsymbol{\alpha}}_i \equiv \boldsymbol{\alpha}_i - \mathbf{A}_i(\mathbf{C}^{(i)} - \mathbf{H}_i^{\#}\mathbf{C}^{(f)})x_{\mathbf{a}}; \quad \tilde{\mathbf{S}}_i = \mathbf{S}_i + \mathbf{A}_i(\mathbf{C}^{(i)} - \mathbf{H}_i^{\#}\mathbf{C}^{(f)})\mathbf{S_a}(\mathbf{C}^{(i)} - \mathbf{H}_i^{\#}\mathbf{C}^{(f)})^T + \mathbf{A}_i\mathbf{C}^{(i)}\mathbf{S_{coin}}\mathbf{C}^{(i)T}, \tag{2}$$

where $\mathbf{C}^{(i)}$ and $\mathbf{C}^{(f)}$ are sampling matrices from the fine to the L2 or fused grid, respectively. The fine grid contains all used grids, i.e. L2 grids and fusion grid. $\mathbf{S}_i$ is the CM of each input product, $\mathbf{S}_{coinc}$ is the coincidence CM, and $\boldsymbol{\alpha}_i$ is originally defined as:

$$\boldsymbol{\alpha}_i = \hat{x}_i - (\mathbf{I} - \mathbf{A}_i)x_{\mathbf{a}i}, \tag{3}$$

$\mathbf{I}$ being the identity matrix. Together with the profile, the complete data fusion provide also the associated CM:

$$\mathbf{S_f} = \Big( \sum_{i=1}^{N} \mathbf{H}_i^{\#T} \tilde{\mathbf{S}}_i^{-1} \mathbf{A}_i \mathbf{H}_i^{\#} + \mathbf{S_a}^{-1} \Big)^{-1}, \tag{4}$$

and the AK:

$$\mathbf{A_f} = \Big( \sum_{i=1}^{N} \mathbf{H}_i^{\#T} \tilde{\mathbf{S}}_i^{-1} \mathbf{A}_i \mathbf{H}_i^{\#} + \mathbf{S_a}^{-1} \Big)^{-1} \sum_{i=1}^{N} \mathbf{H}_i^{\#T} \tilde{\mathbf{S}}_i^{-1} \mathbf{A}_i \mathbf{H}_i^{\#}. \tag{5}$$

As these equations suggest, the performance of the CDF method depends on the quality and consistency of the input profiles. However, it offers several advantages. First, it allows for greater control over the a priori constraint imposed on the fused product, which can theoretically be chosen independently of the a priori constraints of the Level 2 products being merged. Second, it handles profiles on different grids, and allows for fusion onto an arbitrary target grid. By combining information from the Level 2 products while removing their individual constraints, the method enables the generation of a product with potentially higher information content and lower uncertainty. As a diagnostic method for the fusion process compared to each L2 input, the number of degrees of freedom (DOFs), given by the trace of the AK, can serves as a measure of vertical information content. Moreover, the diagonal elements of the AK can be analyzed to assess whether and where vertical resolution increases, and errors can be evaluated as the square root of the diagonal elements of the variance-covariance matrix. Finally, the CDF method



inherently requires consistency among input data, thereby also providing a means for verifying their quality.

**CDF Setup**

Before fusing MIPAS–IASI profile pairs, we harmonized the data through two main steps: (1) State-vector extension. We extended the IASI state vector to include the water continuum and offset parameters present in the MIPAS state vector but not directly retrieved by IASI. Following Tirelli et al., 2021, we appended zero-value entries to IASI's AK and added zeros on the off-diagonal and $10^{-2}$-values on the diagonal outside the profile. This choice reflects the structure of the MIPAS a priori CM and avoids numerical instabilities. (2) Partial-column conversion. We converted MIPAS profile values to moles per unit volume

and then integrated them vertically between consecutive levels using the Levels As Layers (LAL) algorithm on a regular z grid van Peet, 2018. Interpolation matrices were constructed on partial columns, assuming constant density within each layer and computing the fractional overlap between original and new layers. We selected this method after comparing several interpolation schemes.

**CDF Tuning**

| CDF version | Interpolation CM Source | Off-diagonal | Coincidence CM Source | Off-diagonal | k |
|---|---|---|---|---|---|
| v00 | FUSED a priori CM | | Fused a priori CM | | 0.05 |
| v01 | FUSED a priori CM | correlation length | Fused a priori CM | | |
| v02 | A priori L2 | | A priori L2 | | |
| v03 | A priori L2 | | A priori L2 | | |
| v04 | A priori L2 | | A priori L2 | correlation length | |
| v05 | A priori L2 | | Fused a priori CM | | |
| v06 | A priori L2 | | Fused a priori CM | correlation length | |
| v07 | A priori L2 | correlation length | A priori L2 | | 0.025 |
| v08 | A priori L2 | | A priori L2 | correlation length | 0.01 |

**Table 1.** Summary of the nine selected CDF versions based on different choices of interpolation and coincidence covariance matrices (CM), use of off-diagonal reconstruction with a correlation length of 6 km, and application of a factor $k$ on the coincidence CM.

The CDF algorithm offers flexibility in tuning several parameters. One of the strengths of the fusion process lies in the ability to define both the a priori constraint and the vertical grid for the fused product. Prior to performing the fusion, we condicted a set of tests in which we modified both the a priori constraint and the vertical grid of the input datasets. These tests showed that such modifications resulted in stable outputs for MIPAS, while for IASI, they introduced instabilities, likely due to the characteristics

of the corresponding covariance (CM) and averaging kernel (AK) matrices. In this study, we selected the IASI vertical grid as the fusion grid and adopted the IASI a priori constraint for the fused product. This choice is scientifically justified for several reasons. First, since the study focuses on the UTLS and lower troposphere, the IASI grid—by design optimized for these atmospheric regions—offers a natural basis for fusion. Second, as IASI is one of the input datasets, retaining its native



grid ensures compatibility with its associated AKs and CMs, minimizing the introduction of numerical instabilities observed

in prior tests. To ensure consistency, the IASI a priori constraint—composed of both the a priori profile and its associated CM—was extended by including the off-profile elements from the MIPAS a priori state vector. For the corresponding CM, the off-diagonal elements were set to zero, and the diagonal elements outside the profile were assigned the values from the corresponding diagonal elements in the MIPAS a priori CM.

The other two important parameters that can be selected are the interpolation and the coincidence errors. These were used as

the basis for tuning the configuration. We tested nine different configurations combining various approaches to error modeling. These versions are summarized in Table 1. For each version, we applied the CDF algorithm for the year 2008. The table reports the sources of the covariance matrices (CM) used to model coincidence and interpolation errors. When the Fused a priori CM is indicated, the same a priori CM from the fused profiles was used for all inputs to model additional errors. In contrast, 'A priori L2' refers to the use of the individual a priori CM specific to each L2 product. In some versions, off-diagonal elements were

reconstructed by applying a correlation function with a fixed correlation length of 6 km. This reconstruction was implemented using the following formulation:

$$CM_{ij} = \exp\left(-\frac{|z_i - z_j|}{\text{len\_corr}}\right)\sigma_i\sigma_j \tag{6}$$

where $\sigma_i$ and $\sigma_j$ are the standard deviation at levels $i$ and $j$, $z_i$ and $z_j$ are the midpoints of the interpolation grid, and len_corr = 6 km. Conversely, when not specified, CM matrix is considered without modification.

Finally, some versions apply a factor $k$ to the coincidence covariance matrix, effectively scaling it with different strengths.

To identify the most suitable configuration, we considered the results of validation and comparison against L2 products for the year 2008. Metrics, including degrees of freedom (DOFs) for the full profile and below 20 km, total retrieval errors, and the diagonal elements of the averaging kernel matrix (AKM), were evaluated over a large dataset of approximately 7000 profiles.

**2.2.2 Validation Methodology**

Ozonesondes data from WOUDC (World Ozone and Ultraviolet Radiation Data Centre ) stations of 2008 were used to tune the method, while data from the 2009-2012 period were used to validate the entire dataset. Ozonesondes profiles were converted into partial columns, and we assumed a 15% uncertainty for each sonde profile in all our metrics. Moreover, the validation was performed using coincident fused-ozonesonde profiles, after smoothing the ozonosonde profiles with the AK of the fused

profile:

$$x^{smoothed}_{sonde} = x_\mathbf{a} + \mathbf{A}(x_{sonde} - x_\mathbf{a}). \tag{7}$$

where $x_\mathbf{a}$ is the a priori profile, $\mathbf{A}$ is the AK, and $x_{sonde}$ is the ozonosonde profile.

For the tuning phase, only stations between 5°N and 55°N were considered. This latitudinal range provides a sufficient number of coincidences for the fusing step, while avoiding outliers due to large latitudinal differences. Spatial and temporal thresholds



for coincidences between fused profiles and ozonesondes were set at maximum distance of 200 km from the station, and temporal interval of 3 hours before and after the starting and the ending time of the sounding. For each coincident fused-sounding profile, we determined the absolute and percentage bias. The metrics used for the validation are the mean percentage bias over the entire latitudinal band, the mean standard deviation, the mean standard error, and the composite error. This last one should be directly comparable with the standard deviation, and is obtained as the quadratic sum of the noise of the fused

product and the error on the ozonesonde. For the sake of clarity, we summarize the relevant metrics below. The mean percentage bias can be determined as the mean absolute bias divided by the mean ozonesonde profile:

$$< bias_{percent,i} >= 100 \times \frac{< x_{\mathbf{f},i} - x_{sonde,i} >}{< x_{sonde,i} >}, \tag{8}$$

where $<>$ stands for arithmetic mean, and index $i$ refers to each layer. The percentage mean standard deviation is calculated as:

$$< \sigma_{percent,i} >= 100 \times \frac{< \sigma_i >}{< x_{sonde,i} >}, \tag{9}$$

where $< \sigma_i >$ is the mean standard deviation of the $i-th$ layer. The variance deriving from the composite error is the square of:

$$< err_i^{comp} >=< \sqrt{\mathbf{S}_{f,ii}^{noise} + err_{sonde,i}^2} >, \tag{10}$$


where $\mathbf{S}_{f,ii}^{noise}$ is the diagonal noise element of the fused CM, and $err_{sonde,i}$ is the error of the ozonesonde profile. Then the associated percentage error can be obtained as:

$$< err_{percent,i}^{comp} >= 100 \times \frac{< err_i^{comp} >}{< x_{sonde,i} >}. \tag{11}$$


For the validation over the 2009–2011 period, the same criteria and metrics were applied. In this case, ozonesonde stations were grouped into latitudinal bands in addition to the 5°N–55°N band. The aim of this grouping was to investigate the performance of the fused product across different latitudinal zones and to provide a spatial characterization of the dataset.

The choice of the latitudinal bands was driven by the number of profile coincidences available within each band. Due to the

limited number of coincidences, the following broad latitudinal bands were selected to ensure a minimum level of statistical representativeness in each zone: (–75°S to –25°S), (–25°S to 25°N), and (25°N to 75°N).

### 2.2.3   Comparison Methodology

Comparative analyses were carried out to assess the performance of the fused product against each individual Level 2 input dataset, with a specific focus on the tuning performed for the year 2008, and extended more generally to the other years.





We compared the fused product and each L2 dataset with ozonesonde measurements. The procedure was similar to the validation process; the ozonesonde profiles were not smoothed using the individual averaging kernel matrices (AKs) of each instrument, in order to ensure a consistent comparison across the different products. This step was important for assessing the quality of each L2 input dataset and the fused product, and for applying potential corrections if needed.

## 3   Results and Discussion

As previously mentioned, the tuning phase was performed using data from 2008 over the 5°–55°N latitude band. This phase focused on the initial validation of the different versions of the fusion method against ozonesonde profiles, on the analysis of DOFs, and on the retrieval errors of the fused and L2 products.

Figure 2 shows the validation results for the nine proposed versions of the fusion method. All versions exhibit a bias peak around 10 km; however, this peak is reduced (by less than 25%) in version v05. Bias alone is not sufficient to detrmine the

best-performing version. Due to the different constraints applied in each version, some profiles exhibit oscillatory behavior or even negative values. Therefore, the number of profiles containing negative values was also considered as an additional criterion for evaluating the quality of each version. As a result, each version may consequently have a different number of valid profiles. The quantity $n$ in Figure 2 shows, for each version, the number of profiles contributing to the mean. More strongly constrained versions generally exhibit fewer negative values. However, a stronger constraint typically reduces the information

content of the profiles, and thus the number of DOFs. To improve the statistical robustness of the analysis, this aspect together with DOFs was also assessed using a larger dataset, selecting six random days in 2008 and including all fused profiles within the region, rather than only those coincident with ozonesondes.

| CDF version | N tot | N neg (%) | Fused (tot & u20 km) | MIPAS (tot & u20 km) | IASI (tot & u20 km) |
|:---:|:---:|:---:|:---:|:---:|:---:|
| v00 | 7067 | 22.9 | 8.81 (3.50) | 8.09 (2.75) | 3.29 (1.85) |
| v01 | | 0.8 | 8.13 (3.28) | 7.47 (2.53) | |
| v02 | | 0.0 | 6.29 (3.12) | 6.31 (2.45) | |
| v03 | | 0.4 | 7.66 (3.55) | 8.74 (2.96) | |
| v04 | | 1.5 | 7.78 (3.68) | 8.74 (2.96) | |
| v05 | | 11.6 | 9.41 (3.64) | 8.74 (2.96) | |
| v06 | | 10.6 | 9.21 (3.71) | 8.74 (2.96) | |
| v07 | | 4.0 | 8.78 (3.35) | 8.74 (2.96) | |
| v08 | | 0.3 | 7.17 (3.71) | 6.31 (2.45) | |

**Table 2.** DOFs and number of profiles containing negative values analysis. Header is composed of: the total number of fused profiles in 6 days (*N tot*), the percentage of fused profiles with more than one negative value (*N neg*), and the mean number of degrees of freedom both over the full profile and below 20 km, for both the fused (*Fused (tot & u20 km)*)product and for L2 (*MIPAS (tot & u20 km)*; *IASI (tot & u20 km)* products.





**Figure 2.** Validation of the 9 versions of the CDF method in the tuning phase, performed in 2008 within the latitudinal band $5°$ -$55°$ N . The blue line represents the average percentage bias between the fused profiles and the smoothed ozonesondes with the AK of the fused profile. The red dashed line indicates the standard deviation of the mean bias ($2\sigma$), the black dashed line represents the combined error ($2\sigma$), while the orange bars show the standard error of the mean. The background gray profiles represent the $n$ averaged profiles. In the legend, and DOFs mean u20 represents the average number of degrees of freedom below 20 km.





Table 2 presents, for each version, the total number of profiles considered, the percentage of fused profiles containing at least one negative value, and the mean number of DOFs, both over the full vertical range and below 20 km. These metrics are reported for the fused product as well as for the original Level 2 datasets. Since the original Level 2 products are defined on different vertical grids and may rely on different a priori information, their DOFs are not directly comparable. To enable a consistent comparison, MIPAS profiles were interpolated onto the common fusion grid and a shared a priori using the CDF algorithm. This step harmonizes vertical resolution and alignment, reducing discrepancies due to differing retrieval configurations and enabling a meaningful comparison of the DOF values across datasets. Among all versions, v00, followed by v05 and v06, shows the highest percentages of profiles with at least one negative value. Nevertheless, these versions, together with v01, perform best in terms of DOFs and agreement with MIPAS and IASI. This is especially evident for v05, which shows the highest DOFs across the full vertical range and a notable gain in DOFs (+0.67) over MIPAS, while also offering a good compromise in the lower atmosphere (+ 0.68 below 20 km). For MIPAS, values below 5 km are reconstructed using the CDF algorithm, since the instrument does not provide measurements below this altitude. Therefore, the actual improvement in vertical information content should be primarily assessed with respect to IASI, which provides 3.29 degrees of freedom over the full profile, of which 1.85 are below 20 km. These results clearly demonstrate how limb measurements can significantly improve nadir observations by increasing the vertical information content. Notably, the increase in DOFs is observed not only in the altitude range directly sampled by MIPAS, but also in regions where MIPAS does not provide measurements. This is due to the vertical correlations introduced by the fusion process, which propagate information from the limb instrument across the vertical profile.

This outcome is fully consistent with one of the key objectives of our work, namely to exploit the complementary sensitivities of limb and nadir sensors to enhance the vertical information content of the resulting product, especially in the troposphere and in the UTLS. Furthermore, by fusing individual MIPAS and IASI retrievals, we can isolate and assess the specific contribution of each instrument to the enhanced information content, providing a detailed understanding of how the synergy between the two geometries improves the retrieval.

Finally, looking at the fused total errors in Figure 3, an error peak is visible around 10 km. From version v05 onwards (through v08), the errors are lower throughout the entire profile compared to earlier versions.

Based on all the considerations discussed above, version v05 was selected for further analysis, as it offers the most balanced trade-off, especially with regard to bias, degrees of freedom, and retrieval errors.

Version v05 was subsequently applied to the remaining years. Figure 4 shows the comparison between fused and L2 for 2008-2011.

As can be seen, the fused product shows an improvement over the L2 products across all years, starting from the fact that it provides information from the surface to high altitudes. Based on a range-dependent analysis, the following considerations can be drawn.

Above 30 km, the difference with ozonesondes increases, reaching a positive bias exceeding 100%. However, at these altitudes, the accuracy of ozonesonde measurements decreases due to the very low pressure, and this bias should not be considered fully reliable (Stauffer et al., 2014). Between 15 and 30 km, the fused profile shows smaller differences with ozonesondes than MIPAS, and even more so compared to IASI. In this altitude range, the fusion is mostly driven by MIPAS, leading to improved





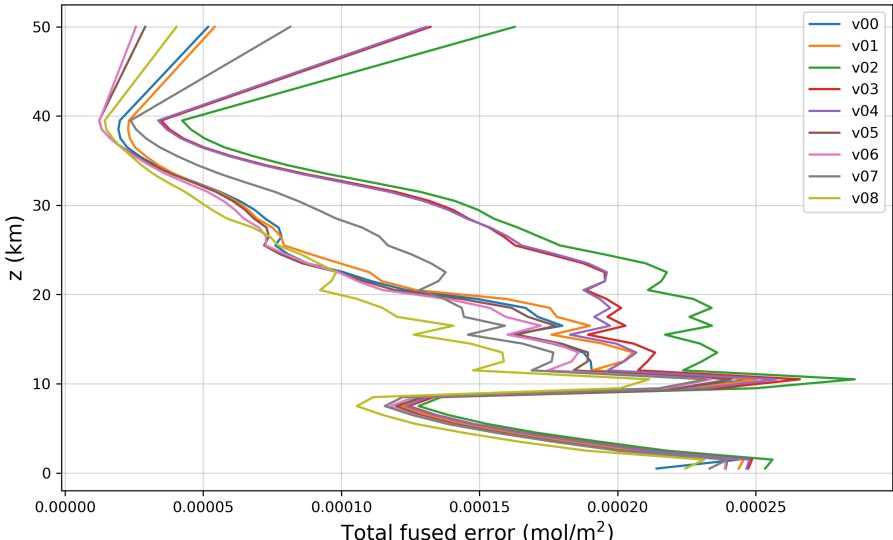

**Figure 3.** Mean total errors of fused products for each CDF version.

performance relative to IASI. Notably, the fused DOFs are equal to those of MIPAS, indicating that the fusion does not increase the retrieval sensitivity in this range. However, the influence of IASI is still evident in the comparison with ozonesondes, indicating a potential impact on the profile shape or bias characteristics. Between 20 and 30 km, MIPAS contributes the largest number of measurements. Below this range, its decreases, as shown in the right panels of Figure 4.

Below 15 km, the contribution of IASI to the fused product increases, and the fused profile tends to follow the IASI profile. Below 5 km, since MIPAS provide no measurements, the information comes almost entirely from IASI. Nonetheless, even below 15 km, the information propagated from higher layers, where MIPAS data is available, improves the IASI-based retrieval. All these aspects become clearer and more quantifiable when examining the DOFs distributed over multiple altitude ranges for the fused and L2 products over years (Table 3).

| Product | 2008 | | | 2009 | | | 2010 | | | 2011 | | |
|---|---|---|---|---|---|---|---|---|---|---|---|---|
| | $\Delta z_1$ | $\Delta z_2$ | $\Delta z_3$ | $\Delta z_1$ | $\Delta z_2$ | $\Delta z_3$ | $\Delta z_1$ | $\Delta z_2$ | $\Delta z_3$ | $\Delta z_1$ | $\Delta z_2$ | $\Delta z_3$ |
| Fused | 0.25 | 3.24 | 2.27 | 0.29 | 3.39 | 2.22 | 0.36 | 3.59 | 2.27 | 0.3 | 3.01 | 2.29 |
| MIPAS | 0.00 | 2.66 | 2.28 | 0.00 | 2.88 | 2.22 | 0.00 | 3.25 | 2.27 | 0 | 2.41 | 2.29 |
| IASI | 0.19 | 1.63 | 0.74 | 0.25 | 1.66 | 0.74 | 0.32 | 1.65 | 0.74 | 0.27 | 1.56 | 0.73 |

**Table 3.** Mean DOFs for different altitude ranges ($\Delta z_1$: 0–5km, $\Delta z_2$: 5–20km, $\Delta z_3$: 20–30km), and for each product and year.

Finally, the comparison between L2 and fused mean total errors (Figure 5) shows that the fused product exhibits lower errors than each L2 product in the UTLS region, and more generally below 18 km in all years. Above 18 km, the fused error is much lower than that of IASI and comparable to that of MIPAS, where MIPAS performs best. The slightly higher fused





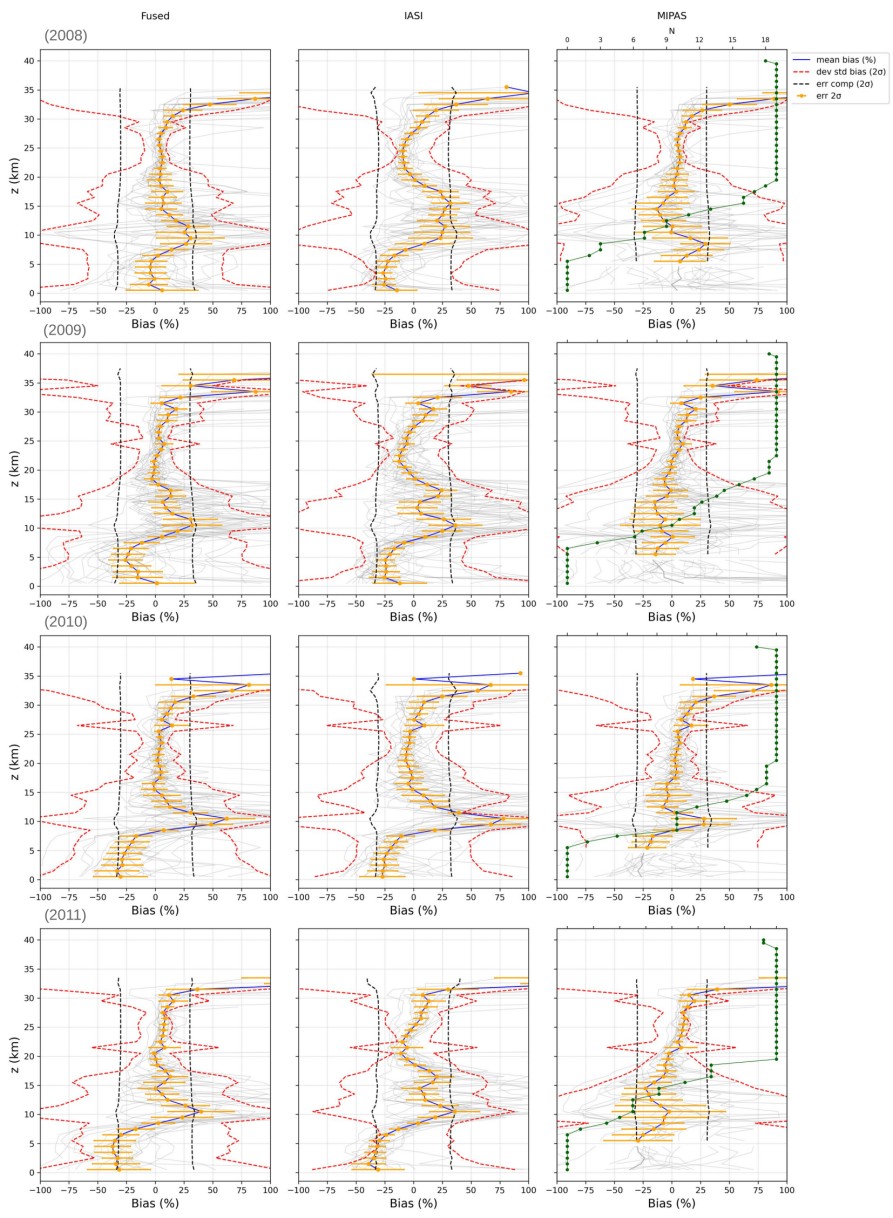

**Figure 4.** Comparison between fused, IASI, MIPAS and ozonosondes for version v05 over all years. The blue line represents the average percentage bias between fused or L2 and ozonesonde, where the ozonesonde has not been smoothed with the AK. The dashed red line represents the standard deviation of the mean at $2\sigma$. The dashed black line represents the combined error ($2\sigma$), while the orange bars represent the standard error of the mean. The transparent gray profiles are the averaged profiles. The green profile represents N, which is the number of MIPAS measurements at each point of the vertical grid.





error compared to MIPAS is likely due to additional uncertainties introduced during CDF processing, such as co-location and interpolation errors.

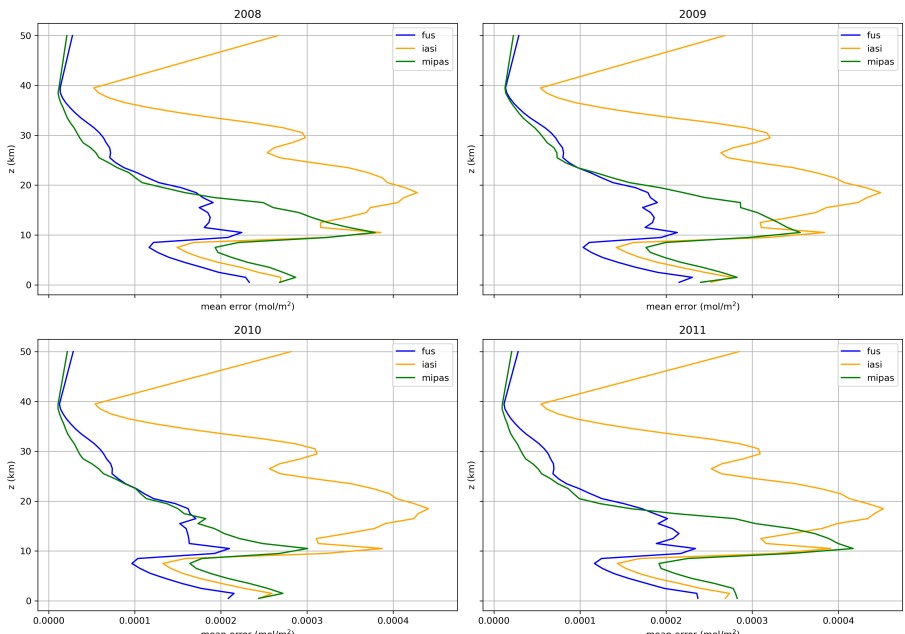

**Figure 5.** Comparison of the mean total errors of v05 fused products and L2 for each year.

Validation over the 5–55°N latitudinal band for the remaining years 2009–2011 confirms the key features of the fused dataset already highlighted previously (Figure 6). The fused product exhibits a significant bias across most altitude levels. Between 10 and 30 km, the bias is positive and always below 25%, peaking around 10 km. Below 8 km, the fused product shows a

significant negative bias, also remaining within 25%.

The validation across three latitudinal bands for the same years (Figure 7) allows assessment of the spatial variability of the bias in the fused product, although this analysis is limited by the number of coincident observations available in each region.

In the latitude band -75° to -25° South, which covers most of the Southern Hemisphere including Antarctica and sub-Antarctic regions, the statistics based on 28 profiles show a positive bias of less than 25% throughout the vertical range, although it is

not significant below 8 km and between 20–30 km. A significant peak of approximately 35% is observed around 10 km.

In the -25° to 25° latitude band, which includes tropical and subtropical regions, the statistics rely on 10 coincidences, and the bias shows much stronger variability. It is negative below 12 km and becomes significant around 10 km, reaching a minimum of approximately -35%. Between 12 km and 18 km, The bias exceeds 100%, becoming very large. Above 18 km, the bias is no longer significant. However, due to the limited number of coincidences, no definitive conclusions can be drawn

In the 25° to 75° North band, representing the Northern Hemisphere (temperate and sub-Arctic regions), the number of coincidences increases to 71. The statistics indicate that the bias is not significant across most of the vertical profile: it is negative



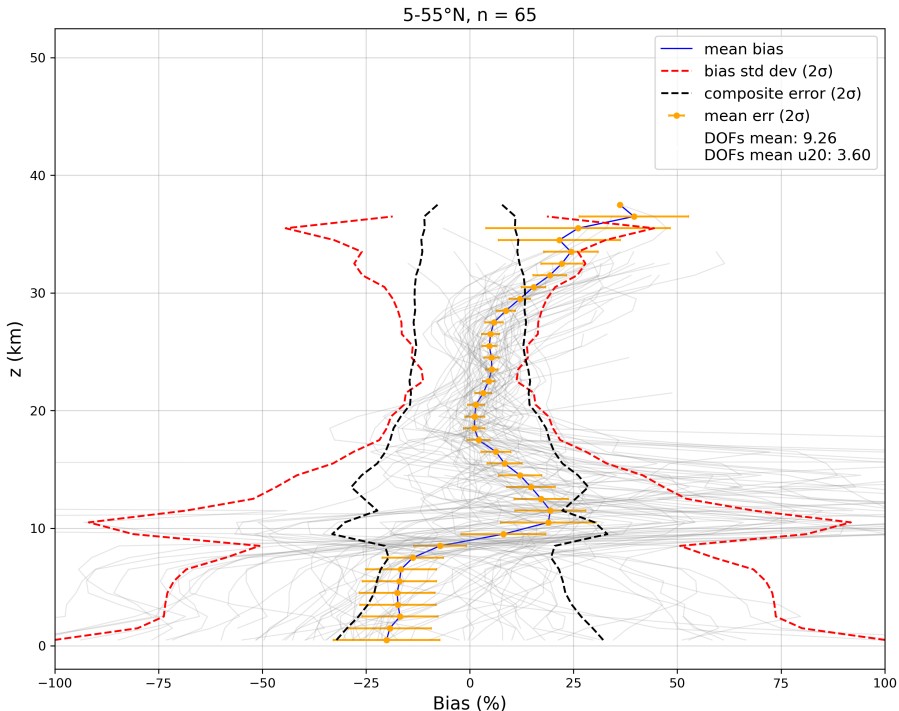

**Figure 6.** Validation of version v05. The blue line shows the mean percentage bias between the fused profiles and the ozonesondes smoothed with the fused product's AK. The red dashed line indicates the standard deviation of the mean bias ($2\sigma$), while the black dashed line represents the combined error ($2\sigma$). The orange bars show the standard error of the mean. The transparent grey lines correspond to the individual averaged profiles. Here $n$ indicates the total number of valid fused profiles considered.

and below 25% below 8 km, and positive above 8 km, with a peak of about 25% near 10 km.

These results suggest that the fused product improves upon the individual products in terms of bias over the entire vertical
profile. However, a consistent feature across all regions is the presence of a bias peak around 10 km, which corresponds to
the altitude range typically associated with stratosphere-to-troposphere intrusion events. This feature appears to originate from
IASI, as can be observed in Figure4 and is also consistent with the findings of Boynard et al., 2016, who reported a similar bias
in the FORLI IASI dataset.
This observation motivated a bias correction of the IASI input profiles prior to applying the CDF algorithm. The fusion algo-
rithm is mathematically robust to such modifications. Therefore, we applied a correction to each IASI profile for the year 2008
by subtracting the percentage bias identified in the IASI validation over the 5–55°N region for 2008.
The new validation for 2008 (Figure 8) shows that, compared to the results presented in Figure2, correcting the IASI bias leads
to an improvement in the fused product.



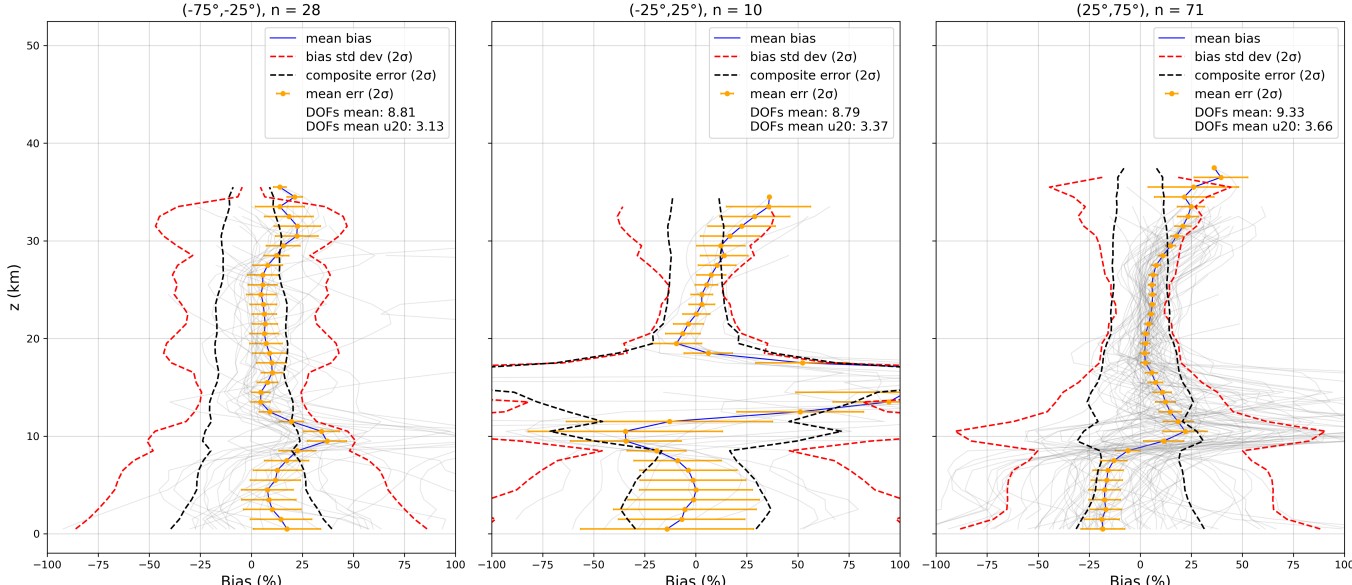

**Figure 7.** Validation of v05 across three latitudinal bands. The blue line represents the mean percentage bias between the fused profiles and the ozonesondes smoothed using the fused product's AK. The red dashed line indicates the standard deviation of the mean bias ($2\sigma$), the black dashed line represents the combined error ($2\sigma$), and the orange bars show the standard error of the mean. The transparent grey lines correspond to the individual averaged profiles. Here $n$ indicates the number of valid fused profiles considered.

The new validation for 2008 (9) shows that correcting the IASI bias significantly improves the fused product: the peak near 10 km is greatly reduced and becomes statistically non-significant. Applying the same IASI bias correction derived for 2008 to the remaining years yields the results shown in Figure 9. It can be seen that the 2008 IASI bias also improves the fused product for the other years, although to varying degrees. This suggests that the IASI bias changes from year to year.

Nevertheless, this correction preserves the key characteristics of the fused product in terms of DOFs and retrieval errors, while considerably reducing the bias—especially in the UTLS region, where it becomes nonsignificant.





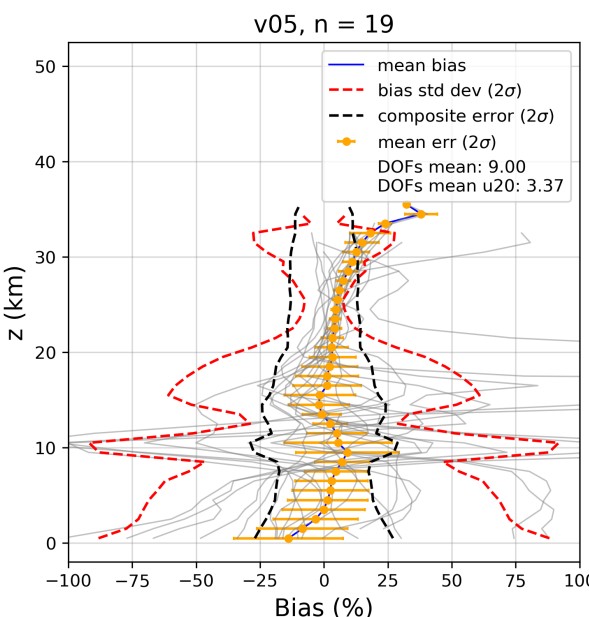

**Figure 8.** Validation of v05 with IASI bias correction applied for 2008. The blue line shows the mean percentage bias between the fused profiles and the ozonesondes smoothed using the fused product's AK. The red dashed line indicates the standard deviation of the mean bias ($2\sigma$), the black dashed line represents the combined error ($2\sigma$), and the orange bars correspond to the standard error of the mean. The transparent grey lines represent the individual averaged profiles. Here $n$ indicates the number of valid fused profiles.

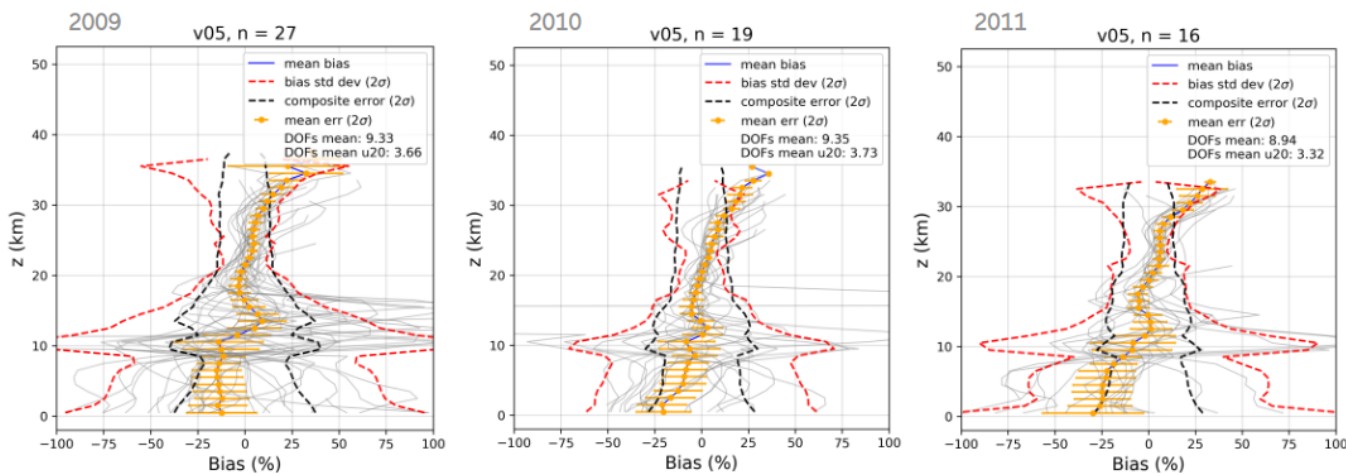

**Figure 9.** Validation of version v05 with IASI bias correction for 2009–2010–2011. In blue is the mean percentage bias between fused profiles and ozonesondes, smoothed with the fused AK. The red dashed line indicates the standard deviation of the mean bias ($2\sigma$), while the black dashed line represents the composite error ($2\sigma$). Orange bars show the standard error of the mean. The semi-transparent grey profiles represent the individual averaged profiles. Here, $n$ denotes the number of valid fused profiles considered.



## 4   Summary and Conclusions

A new ozone dataset has been developed through the CDF algorithm, combining limb observations from the Michelson Inter-
ferometer for Passive Atmospheric Sounding (MIPAS) with nadir observations from the Infrared Atmospheric Sounding Inter-
ferometer (IASI). The primary objective is to support the study of stratospheric intrusions (SIs), which require high-resolution
measurement both vertically and horizontally, especially in the Upper Troposphere–Lower Stratosphere (UTLS). The fusion
targets the overlapping operational period of the two instruments (2008–2011), during which ozone profiles retrieved with
optimal estimation are available, with a particular focus on the Himalayas, a hotspot for SI events. In this context, the aim
of using CDF applied to MIPAS and IASI is to evaluate and enhance the propagation of information from high-quality limb
measurements in the stratosphere down to the troposphere, where the MIPAS sensitivity decreases and retrieval uncertainties
increase, or MIPAS data are unavailable - yet IASI provides coverage.

After tuning the algorithm on data from 2008, the dataset was validated over the remaining years using ozonesondes from
WOUDC stations. The validation was performed both within the 5°S–55°N band and across three latitudinal bands: -75°S to
-25°S, -25°S to +25°N, and +25°N to +75°N. The final fused product is a L2-like dataset composed of ozone profiles in partial
columns, including associated covariance matrices (CM), averaging kernels (AK), and a priori information (profile and CM),
all mapped onto the IASI grid. The IASI a priori information was also used as a constraint in the fusion process.

The best-performing CDF configuration was the one that models the interpolation error using the a priori CM of each input
and the coincidence error using the fused a priori CM. We evaluated the performance of this configuration in terms of bias
against ozonesondes, DOFs, and errors in 2008, and extended the comparison to the entire 2008–2011 period in terms of DOFs
and L2 profile differences. In general, the fused product shows increased information content compared to the individual input
profiles, with improved DOFs and reduced uncertainty. Results show how MIPAS information is transferred to the lower layers
through the application of CDF, also where MIPAS does not measure. This turns out to be in line with the goals of this work.

Validation results over 2009–2011 in the 5°S–55°N band show a statistically significant bias at most altitude levels. A positive
bias (up to  20%) is observed above 8 km, peaking around 10 km, while a negative bias (reaching  –25%) is found below 8
km. A similar behavior is seen in the other latitudinal bands, with notable variations: (1) in 75°S–25°S, the bias is positive
at all altitudes, with a peak at  35% around 10 km; (2) in 25°S–25°N, the bias exhibits strong oscillations but results are not
considered reliable due to a low number of coincidences; (3) in 25°N–75°N, a significant positive bias (25%) is observed above
8 km, while a significant negative bias (–20%) is found below.

In most cases, the fused product shows a consistent positive peak ( 25%) around 10 km. Comparison with the L2 IASI product
suggests that this bias is inherited from IASI, consistent with findings from Boynard et al. (2016). Assessed the robustness
of the fusion algorithm to input bias, we corrected IASI profiles based on the 2008 bias and reran the fusion. This correction
reduced the fused bias to  10%, making it statistically insignificant across most of the profile. Applying the same correction to
the other years showed improvements, though analysis indicates a temporal variability in the IASI bias. This variability—both
temporal and latitudinal—is also supported by Boynard et al. (2016), and warrants further investigation. However, it highlights



the potential to further refine the dataset, particularly in reducing the bias in the UTLS.


The ozone dataset developed in this study provides a new tool for improving the representation of atmospheric ozone. It is worth emphasizing that the potential for assimilating fused data products has been thoroughly investigated within the AURORA project (Advanced Ultraviolet Radiation and Ozone Retrieval for Applications), where numerical experiments using the CDF algorithm demonstrated the technical feasibility of assimilating such products (Cortesi et al., 2018).

This paves the way for further exploring the potential of the above-mentioned dataset of fused ozone profiles for data assimilation in reanalyses such as ERA5 and CAMS, which are known to exhibit biases in both tropospheric and stratospheric ozone due to uncertainties in assimilated observations and limitations in capturing ozone dynamics and chemistry accurately (Inness et al., 2019; Li et al., 2022). Additionally, this dataset can serve as a valuable reference for detecting discrepancies in the physico-chemical modeling of ozone both in reanalyses and in coupled chemistry–climate models.

Finally, this dataset lays the foundation for future improvements in atmospheric reanalyses, modeling, and observational strategies. It also supports air quality studies and contributes to addressing key challenges, such as quantifying the stratospheric contribution to tropospheric ozone, particularly in hotspot regions like the Himalayas.

*Code and data availability.* Data and code used and produced within this study are available from the corresponding authors upon request.

*Author contributions.* All co-authors contributed to the development of the work. Particularly, LG in her PhD developed the study reported
in this paper. Together with NZ she implemented the CDF algorithm coding in all its steps (set up, tuning, and validation). NZ validated OE-reprocessed MIPAS data. SC and CT helped with the theoretical set up of the CDF. Moreover, SC helped with code debugging. MIH together with PR supported the understanding of the context of STE processes and the analysis of results. UC coordinated the research group together with EB. PR provided reprocessed MIPAS data. LG wrote the first draft of the paper, and all the co-authors gave their contribution on the document review.

*Competing interests.* The authors declare that they have no conflict of interest.

*Acknowledgements.* We thank Dr. Cathy Clerbaux for FORLI-processed IASI data, and Dr. Piera Raspollini for reprocessed MIPAS data used in this work as input for the Complete Data Fusion (CDF) algorithm. Moreover, Liliana Guidetti thanks Michaela I. Hegglin for having hosted for three months part of the work at the Institute of Climate and Energy Systems – Stratosphere (ICE-4) of the Forschungszentrum Jülich (FZJ).



*Financial support.* Part of the research activities described in this paper were carried out with contribution of the Next Generation EU funds within the National Recovery and Resilience Plan (PNRR), Mission 4 - Education and Research, Component 2 - From Research to Business (M4C2), Investment Line 3.1 - Strengthening and creation of Research Infrastructures, Project IR0000038 – "Earth Moon Mars (EMM)". EMM is led by INAF in partnership with ASI and CNR.



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
