# Peer review of "Development and Validation of a New Ozone Dataset Using Complete Data Fusion of MIPAS and IASI Observations: A Step Towards Understanding Stratospheric Ozone Intrusions in the Himalayan Region"

_EGUsphere, 2025_

## Referee Comment (RC1)

[General comments]

Basically, this study looks interesting and meaningful because recently the intrusion of stratospheric ozone into the troposphere has been significantly discussed related to the issue of high tropospheric ozone (particularly in Asia). Since vertical ozone profile datasets are very rare, it is time to think about the active usage of satellite ozone profiles. While it has some uncertainty, validated satellite ozone profile can provide useful information to the ozone research community, I believe. In this context, I would like to put some weight on the value of this study. However, there are unclear parts in this manuscript. Thus, sincere responses from authors about suggested comments below are very required and proper revision looks very necessary. In spite of the meaningful topic, data analysis part does not look qualified.

[Specific comments (L: line)]

- Introduction: It is not clear in this manuscript, "Why MIPAS and IASI are used in this study"? There are longer measurements of ozone in both limb (e.g., MLS) and nadir (e.g., OMI and TROPOMI) satellite missions. What can we get distinguished information from MIPAS+IASI combination?

- L47: It is not clear the meaning of "adequate horizontal resolution". Here, is it really necessary for addressing the resolution issue related to the STE?

- L49: Well, it seems that authors would mention 'relatively' higher vertical resolution of limb sounding than nadir-viewing. It is OK, but the question for "Is the vertical resolution of limb sounding is enough for detecting the STE well?" is separated one (3km vertical resolution is very rough actually to see the STM, compared to the ozonesonde measurement). Thus, the 'high vertical resolution' in line 47 looks not proper to me. I do not think that the 'resolution' is the most important motivation for combining limb and nadir viewing satellite data for the STM study. Since limb-sounding does not have the tropospheric profile, combining limb and nadir looks meaningful for the analysis of STM in the UTLS.

- L55-59: Then what is the distinguished motivation of this study compared to suggested previous studies? Especially, I do not know what the main difference is from Tirelli et al. (2025), which seems performed the limb-nadir merging.

- L64: "to evaluate and enhance" => not understanding the meaning of 'enhance' here.

- L67: The full name of 'UTLS' was already suggested in line 46.

- L69-70: Again, then what is the distinguished merit of this study different from Ceccherini (2016), that looks performed the MIPAS + IASI using the CDF method?

- L78: The full name of 'MIPAS' was already suggested above.

- L78: Please correct this sentence to indicate either the launched year, or total duration of MIPAS operation

- L126: It seems necessary to show the portal link here.

- L127: (FORLI-O3) => (FORLI)

- L227: no need to show full name of CM once more again here.

- L241-242: What is the meaning for "to tune the method"? How is it associated with Table 1? What is it tuned, why is it needed, and is it a reliable approach? Although it is OK, is it enough only with ozonesonde data in 2008, just a specific one year?

- L249: Is there any previous work showing that the latitudinal difference of ozonesonde

measurements affect the validation result? It seems that some references are needed related to the statement "while avoiding outliers due to large latitudinal differences".

- L271: The statement "For the validation over the 2009–2011 period," here does not look connected to the statement "data from the 2009-2012 period were used to validate the entire dataset" in L242.

- L276: Why here the ozonesonde data from -75 S to 75 N are used, different from the usage of data from 5 to 55 N above. I cannot get the explanation in this last paragraph.

- L278-283: What is the difference between the validation with ozonesonde (2009-2011) and comparative analyses with ozonesonde (2008)? It is not clear to me.

- L288-289: Why does the peak bias occur around 10 km height?

- L291 (and in this part entirely): What does the negative value mean? negative bias or negative ozone? The meaning of negative value, and the reason to consider this negative value should be better described in this part.

- L298-319: This paragraph is too long. Please make two paragraphs.

- L320-328: In contrast, here are 4 paragraphs having short statements. Proper length of each paragraph and smooth connection (flow) of all paragraphs are important for readers' understanding. Please revise this part carefully.

- L320: Again, there is no explanation of high error around 10 km. The reason should be explained. Also, I am not sure if the word 'bias' and 'error' is differently used or has same meaning. It is better to use consistent and accurate wording.

- Chapter 3: Most of statements are just reading figures, not including 'real discussion'. In other words, it is difficult to find the reason why errors are high at certain altitudes.

---

## Author Comment (AC1)

We are grateful to the reviewer for providing comments and suggestions. In the following, we report our response in blue.

While the manuscript is relatively well rounded, a few suggestions could further enhance its impact. First, the dataset covers a relatively short period (2008–2011), which may limit its utility for chemical modelers for the chemistry/dynamics related evaluations. To better showcase the new data sets unique value, I strongly recommend including a specific case study—such as a well-documented stratospheric intrusion event—where the fused product demonstrates improved agreement with ozone sonde data compared to reanalyses like CAMS or ERA5. Additionally, given the manuscript's focus on Stratosphere-Troposphere Exchange (STE) in the Himalayan region, it is noteworthy that the ozone sonde stations shown in Figure 1 are predominantly located in Europe and the United States. Incorporating independent ozone sonde data from the Himalayan region would allow for a more targeted evaluation of the fused dataset in its primary area of interest.

We thank the reviewer for these valuable suggestions.

We agree that the dataset covers a relatively short period, as also noted by Reviewer 1. The temporal extent of the dataset is constrained by the overlap between MIPAS and IASI. As explained to the first reviewer, there are instruments that provide longer time series. However, as we also pointed out in the first comment, while other limb+nadir combinations such as MLS+OMI or MLS+TROPOMI are theoretically possible, they involve greater complexity. At present, we can apply the CDF approach only when all the necessary information (profiles, a priori, CM, and AKM) is available. This requirement is met when products are retrieved using Optimal Estimation (OE) and the full CM and AKM are provided for each profile. We are currently investigating the possibility of applying CDF to products that are not retrieved with OE.

We also acknowledge that the Himalayan region suffers from lack of reference measurements such as ozonesonde profiles or lidar observations. Only a few stations provide vertical ozone profiles, and none of them coincide with the spatiotemporal coverage of our fused dataset, to our knowledge. For this reason, it

was not possible to include independent ozonesonde data from the Himalayan region in the analysis.

This limitation actually highlights the relevance of remote measurements and related analyses based on them. In this context, the fused product can provide data comparable to direct observations while enhancing their value, as it combines complementary information from nadir and limb sounding geometries. Unlike reanalyses, which blend assimilated observations-ERA5 and CAMS assimilate IASI and MIPAS, respectively-with dynamical and chemical forecast models, the fused product is a direct, measurement-based dataset that more closely represents the truth, especially in data-sparse regions. This makes it particularly valuable in areas such as the Himalayas, where observational coverage from both ground stations and remote sensing is extremely limited.

To our knowledge, well-documented case studies of stratospheric ozone intrusion events in the Himalayan region do exist in the literature, but we did not find any within our period of interest (2008–2011). However, thanks to a dataset provided by Dr. Paolo Cristofanelli and Dr. Davide Putero, who worked with the NCO-P station data, we identified a case study for 28 January–2 February 2008, and we have published on Zenodo (https://zenodo.org/records/17590008 ) the fused dataset for these six event days to very partially meet the reviewer's request.

We are currently working on the dataset to develop a methodology that demonstrates the added value of the fused product. While a comparison with reanalyses is certainly of interest, presenting such an analysis in the absence of a thorough investigation of the  requested case study would require more extensive and dedicated treatment. This level of detail would go beyond the scope of the current manuscript, which is highly technical and focused on dataset production and demonstrating its validation and potential applicability.

Finally, I strongly encourage the authors to adopt an open-data policy. Making the dataset publicly available would facilitate broader evaluation and application by the research community, thereby maximizing the impact of this important work. The current statement that data is available upon request may limit its accessibility and utility.

Regarding the adoption of an open-data policy, we fully support this approach and are pleased to share the datasets used in this work and make them available to the scientific community. Specifically, we have published on Zenodo (https://zenodo.org/records/17596981 ) the fused dataset corresponding to the coincidences with ozonesondes, which were employed for the validation of the fused product. We have also published on Zenodo the MIPAS data reprocessed with OE and interpolated onto the IASI vertical grid, used for validation (https://zenodo.org/records/17637275 ), along with supplementary materials (https://zenodo.org/records/17641663 ), such as the ozonesondes used and more detailed data.  In addition, we have made available (https://zenodo.org/records/17590008 ) a fused dataset for six days of stratospheric intrusion event (28 January–2 February 2008) over an extended region (5–55°N; 50–110°E) encompassing the Himalayas, which can serve as a case study. Please note that these datasets are provided in research-specific formats and may not fully comply with standard community conventions. They are intended to enable replication of validation and case study analysis. We have therefore updated the Data and Code Availability section accordingly.

In parallel, we are working to make the complete global dataset, covering the four years of analysis, publicly available. This effort is part of the EMM (Earth Moon Mars) PNRR-funded project mentioned in the financial support section, within which a dedicated platform is being developed to host and distribute these datasets. Our goal is to ensure that the data are fully compliant with Level-2 standards and presented in a user-friendly format, facilitating and ensuring reuse and integration by the scientific community.

---

## Author Response (AR2)

We are grateful to the editor for providing comments and suggestions that ultimately improved the quality and clarity of our work. In the following, we report the specific comments received by the editor followed by our point-by-point responses in blue colour. Accordingly, we have revised the already submitted revised manuscript providing version with modifications in track-change mode (with deletions marked as red and additions marked as blue)

Thank you for your response and revision to the manuscript "Development and Validation of a New Ozone Dataset Using Complete Data Fusion of MIPAS and IASI Observations: A Step Towards Understanding Stratospheric Ozone Intrusions in the Himalayan Region". Based on my review of the revised manuscript, reviewers' comments, author response, and my own reading, the manuscript requires further (minor) revisions as listed below:

1) As commented by one of the reviewers, I feel that the second part of the title can be misleading as there don't seem to be any specific case studies over the Himalayan Region. If additional studies are not feasible, you may consider modifying the title to reflect the contents of the paper. Potential modification could look like "Development and Validation of a New Ozone Dataset Using Complete Data Fusion of MIPAS and IASI Observations: A Step Towards Understanding Stratospheric Ozone Intrusions".

Thank you for your valuable suggestion. Our initial intention was to provide the broader context in which the dataset was developed and intended to being exploited further; however, we fully understand your point and that previously raised by one of the reviewers. We have therefore modified the title as suggested.

2) Please increase font size for most figures as the contents on the x- and y-axis, title, and legend are too small to read.
Thank you. Accordingly, we modified Figures 3-4-5-6-7 increasing the font sizes on the axes, title and legend.

3) Please define all acronyms in their first use in the abstract and main body and ensure that they are not re-defined in multiple places. I found several instances of

re-defining the acronyms in the main body. On lines 41-42, it is better to spell out the name instead of symbol for ozone tracers.

Thank you for your suggestions. We revised all acronyms and their definition throughout the text. Specifically:

- MIPAS: abstract, line 410
- IASI: abstract, line 411
- FR, OR: line 108
- UTLS: line 413
- AK: abstract, lines 182-222-245-292-422
- CM: abstract, lines 182-201-222-241-422
- DOFs: caption Table 2, caption Figure 2, lines 243-332-348
- OE: line 413

Moreover, we revised lines 41-42 spelling out the ozone tracers names.

4) Word separation missing or attention needed for line 58, 213, 308, 389, 397, 402, and Table 2 table caption.

Thank you. We revised these lines according to the comment.

5) Line 181 and Eqn 1: It is unclear what xj is. Could it be a typo for xf?
Thank you for your observation. indeed this was a typo. We revised this line writing xf.

6) Line 188 and Eqn 2: Scoinc might be a typo for Sconc.

Thank you for your observation. 'Scoinc' is a typo here for Scoin. We changed deleting the c.

7) Line 275, Eqn 10: Could you please confirm that the first term on the right is correct? Should it be squared?

Thank you for pointing this out. The first term on the right-hand side, $S_{f,ii}^{noise}$, represents the diagonal elements of the noise covariance matrix, which are

variances. Therefore, the square root is applied to the sum of two variances and is needed.

8) Lines 206-215: You may want to reformat with a bullet form for an easier read.

Thank you for your suggestion. Accordingly, we revised the sentences adding bullet points for better clarity.

9) Table 2 and 3: Should 4-6 columns in Table 2 and 2-5 columns in Table 3 say DOF as those columns are showing DOFs?

Thank you for this observation. These columns indeed show DOFs. To provide clearer information, we have added 'DOFs' to the table headers in both tables.

10) Line 361: "Its decreases" -> "It decreases".
Thank you, we corrected the wording accordingly.

11) Line 439: "Assessed" -> "Assessing"

Thank you, we corrected the wording accordingly.